# Intestinal Mucosa-Associated Lymphoid Tissue Lymphoma Transforming into Diffuse Large B-Cell Lymphoma in a Young Adult Patient with Neurofibromatosis Type 1: A Case Report

**DOI:** 10.3390/medicina58121830

**Published:** 2022-12-12

**Authors:** Hideki Kosako, Yusuke Yamashita, Ken Tanaka, Hiroyuki Mishima, Ryuta Iwamoto, Akira Kinoshita, Shin-ichi Murata, Koichi Ohshima, Koh-ichiro Yoshiura, Takashi Sonoki, Shinobu Tamura

**Affiliations:** 1Department of Hematology/Oncology, Wakayama Medical University, Wakayama 6418509, Japan; 2Department of Pathology, Kurume University School of Medicine, Fukuoka 8300011, Japan; 3Department of Human Genetics, Atomic Bomb Disease Institute, Nagasaki University, Nagasaki 8528523, Japan; 4Department of Diagnostic Pathology, Wakayama Medical University, Wakayama 6418509, Japan; 5Department of Emergency and Intensive Care Medicine, Wakayama Medical University, Wakayama 6418509, Japan

**Keywords:** neurofibromatosis type 1, mucosa-associated lymphoid tissue lymphoma, histological transformation, IgA, whole-exome sequencing, A20

## Abstract

*Background*: Neurofibromatosis type 1 (NF1) is a hereditary cancer syndrome characterized by multiple café-au-lait macules on the skin. Lymphoproliferative malignancies associated with NF1 are limited, although the most common are brain tumors. *Case presentation*: A 22-year-old woman with NF1 was admitted due to abdominal pain and bloody diarrhea. Her laboratory data exhibited macrocytic anemia and elevated IgA levels. Image studies showed diffuse increased wall thickening in the transverse and descending colon without lymphadenopathy and hepatosplenomegaly. A colonoscopy revealed a hemorrhagic ulcerated mass. Pathological analysis of the tumor tissues confirmed IgA-expressing mucosa-associated lymphoid tissue (MALT) lymphoma with histological transformation. Moreover, whole-exome sequencing in tumor tissues and peripheral blood mononuclear cells identified a somatic frameshift mutation of the *A20* gene, which represents the loss of function. The patient responded well to R-CHOP chemotherapy, but the disease relapsed after 1 year, resulting in a lethal outcome. *Conclusions*: MALT lymphoma in children and young adults is extremely rare and is possibly caused by acquired genetic changes. This case suggests a novel association between hereditary cancer syndrome and early-onset MALT lymphoma.

## 1. Introduction

Mucosa-associated lymphoid tissue (MALT) lymphoma is an extranodal marginal zone lymphoma that arises from B cells in the marginal zone of secondary lymphoid tissues [1,2,3]. It is a relatively uncommon clonal B-cell neoplasm that accounts for only 5–8% of all non-Hodgkin lymphoma cases and is most commonly associated with chronic inflammation caused by infections and autoimmune conditions [1,2,3]. MALT lymphoma occurs in various organs, primarily the stomach, followed by the ocular adnexa, lungs, salivary glands, colorectum, and small intestine [4]. In general, localized MALT lymphoma displays an indolent course and has a good prognosis [1,2,3,5]. Accordingly, watchful waiting is recommended in clinical practice for most asymptomatic patients. For patients with *Helicobacter pylori*-positive primary gastric MALT lymphoma, the standard treatment is eradication therapy [6]. The recommended treatment strategy for advanced MALT lymphoma is the same as for advanced follicular lymphoma [1,2,3]. Approximately 2% of MALT lymphomas transform into diffuse large B-cell lymphoma (DLBCL) [5]. Histological transformation results in a worse prognosis [7,8,9].

Neurofibromatosis type 1 (NF1), previously known as von Recklinghausen disease, is a hereditary cancer syndrome characterized by neurofibromas, multiple café-au-lait macules (CALMs), and optic gliomas, with a frequency of 1/3000–4000 persons [10,11,12,13]. Although most patients with NF1 have a relatively favorable prognosis, concurrent high-grade tumors often develop, most commonly, brain tumors [14,15]. Therefore, careful and regular follow-up is necessary for the early detection of malignancy in these patients. In contrast to solid tumors, the frequency of lymphoproliferative malignancies associated with NF1 is controversial. In an English cohort, patients with NF1 were three times more likely to develop lymphoproliferative malignancies compared with healthy subjects [15]. However, in recent studies in Finnish and French cohorts, patients with NF1 showed no increased tendency to develop lymphoproliferative malignancies in their lifetime [16,17]. Moreover, management and treatment strategies for patients with NF1 with lymphoproliferative malignancies have not been established yet.

In the present study, we report the first case of a young adult patient with NF1 who developed MALT lymphoma with histological transformation to DLBCL. Whole-exome sequencing (WES) identified a somatic mutation in the *A20* gene (also called *TNFAIP3*), a key regulator of nuclear factor kappa-light-chain-enhancer of activated B cells, in tumor tissues, and peripheral blood mononuclear cells (PBMCs) [18].

## 2. Case Report

A 22-year-old Japanese woman was referred to our hospital due to unbearable abdominal pain and continuous bloody diarrhea, which had initially presented as recurrent abdominal pain and watery diarrhea 3 months previously. She was diagnosed with NF1 in infancy because of multiple CALMs (more than six) and axillary freckling. In addition, she had a history of epilepsy, mental retardation, and Addison’s disease, and had been taking two antiepileptic drugs (valproate and carbamazepine) and low-dose oral corticosteroids for a long time. There was no family history of consanguineous marriage or genetic disorders.

Physical examination revealed short stature (−3SD), low body weight (−2SD), pale conjunctivae, multiple CALMs on her limbs and trunk (Figure 1), axillary freckling, and tenderness in the left upper abdomen. The liver and spleen were unpalpable, Murphy’s sign was negative, and bowel sounds were decreased. Her vital signs including body temperature and blood pressure were normal. Laboratory findings revealed macrocytic anemia (hemoglobin, 9.1 g/dL; mean corpuscular volume, 117.8 fL), increased IgA level (2072 mg/dL), and decreased IgG (308 mg/dL) and IgM (17 mg/dL) levels (Table 1). No detectable lymphoma cells were found in the peripheral blood at any time. Contrast-enhanced computed tomography (CT) showed marked diffuse wall thickening of the transverse and descending colon (Figure 2a,c), a mass-like lesion in the splenic flexure (Figure 2a,c, arrows), and ascites in the rectouterine pouch (Figure 2b), while lymphadenopathy and hepatosplenomegaly were not observed.

The patient was admitted to our hospital urgently due to severe symptoms. A colonoscopy performed the day after admission revealed mucosal redness, edema, and a hemorrhagic ulcerated mass extending from the splenic flexure to the descending colon (Figure 2d,e). A biopsy of the tumor tissues was performed, and the histological examination revealed diffuse lymphoid infiltration with Dutcher bodies in the lamina propria of the colon (Figure 3a,b). The predominant population of the proliferating cells was medium-sized lymphocytes, but large lymphocytes were also observed in some areas (Figure 3a,b). The majority of medium- to large-sized lymphocytes were positive for CD20, CD79a, IgA, and lambda (Figure 3c–f), but negative for Epstein–Barr virus-encoded small RNA 1 (Figure 3g). The MIB-1 labeling index was 90% in the large cell area (Figure 3h). Based on the cell morphology and immunohistochemical studies, the medium-sized cell regions with a low MIB-1 labeling index were diagnosed as intestinal MALT lymphoma, while the large-sized cell regions with a high MIB-1 labeling index were diagnosed as histological transformation into DLBCL. There was evidence of IgA-expressing lymphoma cells scattered throughout the bone marrow. Cytogenetic analysis of the bone marrow showed a normal karyotype; however, we could not obtain cytogenetic results from the tumor tissues because of poor proliferation.

To identify genetic variants in individual and lymphoma cells, WES and Sanger sequencing were performed after obtaining written informed consent from the patient and her parents, and approval by the Institutional Review Board of Wakayama Medical University (approval number: 57) following the Declaration of Helsinki. Briefly, genomic DNA was extracted from the PBMCs of our patient and her parents, and from the resected colonic tissues of the patient, without fixation [19,20]. WES (at >200 × depth) was performed for four samples using the Illumina MiSeq platform (San Diego, CA, USA). WES revealed three frameshift mutations (in the *A20*, *TNIP1*, and *FAS* genes) and three non-synonymous mutations (in the *TP53*, *BTG1*, and *BCL7A* genes) in the tumor tissues. These were de novo somatic mutations associated with the development of B-cell lymphoma. Moreover, these frameshift mutations were also identified in the patient’s PBMCs. Sanger sequencing identified a frameshift insertion in the *A20* gene, indicating the loss of function in the tumor tissues and PBMCs of our patient (Figure 4). However, a mosaic mutation with low frequency was suspected in the patient’s PBMCs. Therefore, PCR amplicon-based next-generation sequencing (NGS) was performed, which revealed a somatic mosaic (12–15%) of the frameshift mutation in the *A20* gene in the patient’s PBMCs. On the other hand, neither *FAS* nor *TNIP1* gene mutations were identified by PCR amplicon-based NGS.

Considering the risk of secondary malignancies, we selected at least six courses of R-CHOP (rituximab, cyclophosphamide, doxorubicin, vincristine, prednisolone) as standard therapy for DLBCL, but not followed by radiation therapy or upfront autologous transplantation. After starting one cycle of R-CHOP, the patient had no grade 3/4 non-hematological toxicities, and her symptoms improved dramatically. After six cycles of R-CHOP, there were no visible massive lesions on CT and the colonoscopy showed normal findings; however, the follow-up biopsy specimen revealed residual MALT lymphoma. One year after therapy completion, the disease relapsed and the patient’s condition rapidly deteriorated to multiple organ failure, and eventually, a lethal outcome.

We retrospectively reviewed the patient’s medical records to estimate the onset of the lymphoma. IgA levels were initially elevated at the age of 8 years and gradually increased from the age of 16 years, whereas the IgG levels simultaneously decreased (Figure 5a). A CT scan obtained at the age of 18 years showed an infiltrative shadow with a cavity in the right upper lung lobe (Figure 5b, arrow). A transbronchial lung biopsy was performed, and the abnormal shadow was suspected to be due to the proliferation of plasma cell-like tumor cells. However, our patient and her parents declined further examinations. On admission to our hospital, the abnormal lung shadow was still present but disappeared after chemotherapy. These findings suggested that our patient had a history of pulmonary MALT lymphoma.

## 3. Discussion

The *NF1* gene encodes neurofibromin, a negative regulator of the *RAS* proto-oncogene, which affects the RAS/MAPK pathway to promote cell proliferation and survival [10,11,12,13]. Germline loss-of-function mutations in this gene lead to the clinical characteristics of NF1 and increase the risk of solid malignant tumors, juvenile myelomonocytic leukemia, and plexiform neurofibromas [14,15,21]. Furthermore, somatic mutations in the *NF1* gene have been largely linked to some cancers including melanoma, lung cancer, and breast cancer [22]. Among the hematological malignancies, these somatic mutations have been identified with a low frequency in acute leukemia [22]. As for lymphomas in patients with NF1, 44 cases have been reported; five cases of Hodgkin lymphoma and 39 cases of non-Hodgkin lymphoma (18 of B cell, 15 of T cell, and six unclassified) [23]. However, the lifetime risk of developing lymphoma in patients with NF1 is controversial, and somatic mutations of the *NF1* gene in lymphoma are limited [15,16,17]. Genetic testing has been not mandatory to diagnose NF1 [24,25]. Our patient was diagnosed with NF1 by the age of 1 year due to multiple CALMs and axillary freckling during childhood. Although the WES of our patient’s PBMCs did not identify *NF1* genetic mutations, our patient met the revised 2021 diagnostic criteria for NF1 [24,25].

Multiple CALMs were associated with several inherited genetic syndromes such as NF1 and the McCune–Albright, Noonan, Watson, Legius, and constitutional mismatch repair deficiency (CMMRD) syndromes [26]. CMMRD is a rare childhood cancer predisposition syndrome resulting from biallelic germline mutations in DNA mismatch repair genes including *MLH1*, *MSH2*, *MSH6*, and *PMS2* [27]. Most patients with CMMRD syndrome present with multiple CALMs resembling NF1 and have an increased risk of developing hematological, intestinal, and brain malignancies [27,28]. Notably, the most common hematological malignancy in CMMRD syndrome is non-Hodgkin lymphoma [28,29,30]. The WES of our patient’s PBMCs and tumor tissue showed a parental mutation of the *PMS2* gene, but not a biallelic mutation. In this report, we could not establish a definitive diagnosis of CMMRD syndrome in our patient, who had already been diagnosed with NF1. Treatment of cancers in CMMRD is commonly challenging because of the resistance to chemotherapeutics [27,28,29,30]. Further analyses including microsatellite instability, additional immunohistochemistry, and deep sequencing might be helpful not only for the definitive diagnosis of hereditary cancer syndrome, but also for the management of early-onset lymphoma, as in this case [27,28,29,30].

MALT lymphomas are often associated with underlying autoimmune disease and chronic viral infection [1,2,3]. The close examination at the first hospitalization revealed no evidence of these diseases in our patient. *Helicobacter pylori* infection is the major risk factor associated with the development of gastric MALT lymphoma; however, the pathogenesis of MALT lymphoma of the colon remains unclear [31]. The WES of the genomic DNA isolated from the tumor tissues and PBMCs of our patient revealed a de novo frameshift mutation in the *A20* gene, which represents the functional loss of the A20 protein. In particular, the *A20* genetic mutation allele frequency in PBMCs was identified as 12–15%, supporting the somatic mosaic mutation. Mutations and deletions in the *A20* gene are involved in the pathogenesis of MALT lymphoma [32]. Moreover, A20 haploinsufficiency causes early-onset autoinflammatory diseases such as Behçet’s disease [33]. Although the main symptoms of Behçet’s disease were not present in our case, it is possible that the somatic mosaic mutation in the *A20* gene caused chronic inflammatory or autoimmune conditions in the colon, leading to the accumulation of multiple mutations and the development of MALT lymphoma. Furthermore, mutations in other genes including *TP53*, *BTG1*, and *BCL7A* were also confirmed in the tumor tissues, possibly causing the development of DLBCL. Our patient with suspected CMMRD syndrome may have been predisposed to acquire the *A20* genetic mutation. Further analysis of the *A20* somatic mutation identified in our patient may clarify the mechanisms of pathogenesis associated with early-onset non-Hodgkin lymphoma, particularly MALT lymphoma.

Although monoclonal gammopathy was found in 15–20% of patients with low-grade and aggressive B-cell lymphoma, the frequency of IgA-expressing lymphomas was very low [34]. The previous retrospective analysis has also described that one out of 19 patients (5%) with MALT lymphoma had IgA monoclonal gammopathy [35]. Moreover, we found only one case report of histological transformation in IgA-expressing lymphoma [36]. These findings support that our patient with histological transformation in IgA-expressing MALT lymphoma was extremely rare.

Immunoproliferative small intestinal disease (IPSID) is a rare variant of MALT lymphoma with villous atrophy in the small intestine, most commonly described as Mediterranean lymphoma (also known as α heavy chain disease) [37,38,39]. IPSID affects older children and young adults (range, 10–35 years; mean, 25–30 years), with a higher prevalence among those with a low socioeconomic status in developing countries [37,38,39]. Therefore, IPSID should be considered in the differential diagnosis of children and young adults with intestinal MALT lymphoma. The histological features of IPSID are typically characterized by plasma cell infiltration in the gastrointestinal wall, mainly the intestine, and secretion of a monotypic immunoglobulin α heavy chain (the M component is generally IgA) with a lack of the associated light chain [37,38,39]. While our patient was diagnosed with IgA-secreting MALT lymphoma in young adulthood, the onset of the disease was possibly earlier based on the change in her serum IgA levels and the lung parenchymal lesion. Therefore, we also initially suspected IPSID in addition to colorectal MALT lymphoma. However, the pathological analyses showed that the lymphoma cells mainly infiltrated the colon and predominantly expressed a light chain lambda as well as a heavy chain IgA. These findings confirmed the diagnosis of colorectal MALT lymphoma but not IPSID in this patient with NF1.

## 4. Conclusions

We present the first case of a young adult patient with NF1 who developed intestinal MALT lymphoma with histological transformation. In addition, we identified a somatic mutation in the *A20* gene, possibly involved in the development of MALT lymphoma, in tumor tissues, and PBMCs. This case supports that hereditary cancer syndrome should be considered as a differential diagnosis in early-onset MALT lymphoma. Furthermore, identification of the underlying genetic changes in patients with NF1 will enable the pathogenesis and long-term management of secondary malignancies.

## Figures and Tables

**Figure 1 medicina-58-01830-f001:**
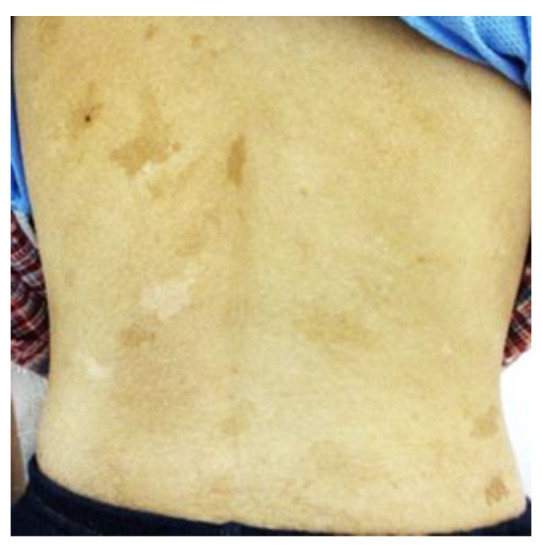
Multiple café-au-lait macules on the back of our patient.

**Figure 2 medicina-58-01830-f002:**
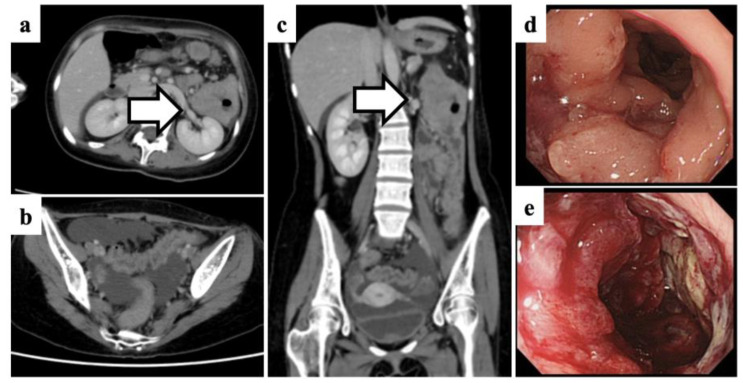
Imaging findings of our patient on admission. Contrast-enhanced computed tomography axial images of the abdomen (**a**) and small pelvis (**b**) and coronal images (**c**) showing circumferential thickening of the colon wall, a mass-like lesion in the splenic flexure (arrows), and ascites in the rectouterine pouch. Colonoscopic images of the lymphoma at the splenic flexure (**d**) and descending colon (**e**).

**Figure 3 medicina-58-01830-f003:**
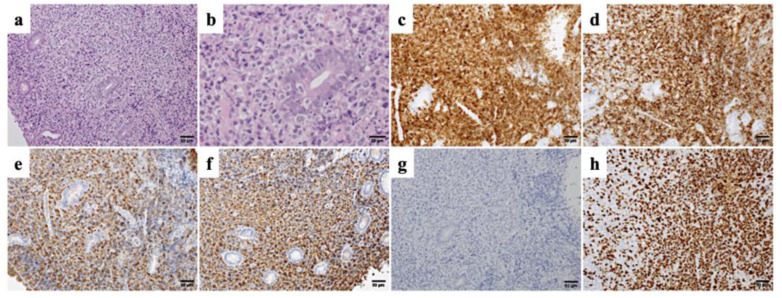
Histopathological findings from the biopsy specimen from the splenic flexure mass. (**a**) Diffuse lymphoid infiltration with Dutcher bodies in the lamina propria (hematoxylin and eosin stain, ×200). (**b**) Higher magnification of the area infiltrated by large-sized lymphocytes (hematoxylin and eosin stain, ×400). (**c**–**h**) Immunostaining results (all ×200): infiltrated lymphocytes were positive for CD79a (**c**), CD20 (**d**), IgA (**e**), and lambda (**f**), and negative for Epstein–Barr virus-encoded small RNA 1 (**g**). The MIB-1 labeling index was 90% in the large-sized cell area (**h**).

**Figure 4 medicina-58-01830-f004:**
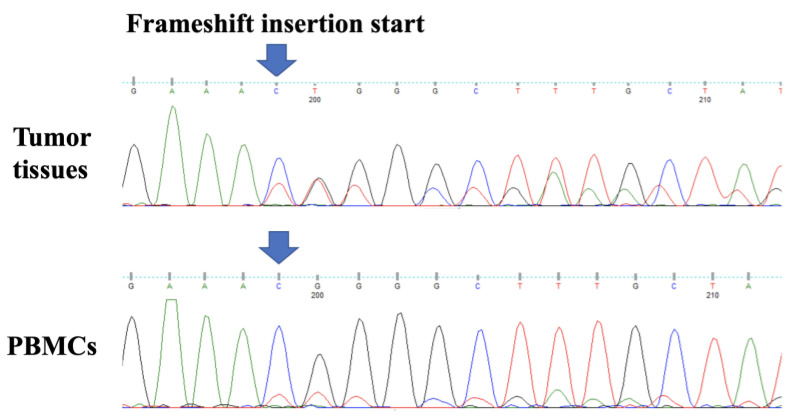
Sanger sequencing electropherogram confirming a frameshift deletion and insertion of the *A20* gene (c.464_467delinsTTTGCTGAAATTTGTTGAAATTTGTTGAAATTT, p.T155Ifs*2). The arrow indicates the position of c.464C. PBMCs, peripheral blood mononuclear cells.

**Figure 5 medicina-58-01830-f005:**
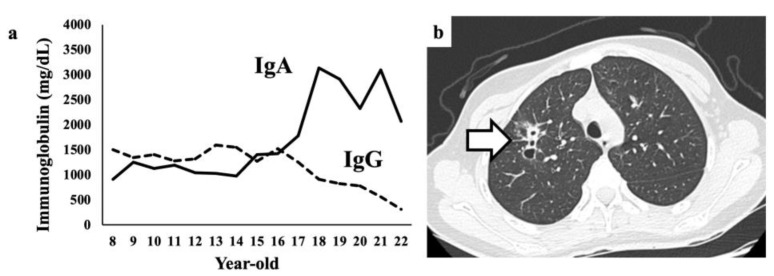
Review of the previous findings in our patients. (**a**) Changes in serum immunoglobulin levels, including IgA and IgG, over time. (**b**) Chest computed tomography image obtained at the age of 18 years (pulmonary window setting) showing an infiltrative shadow with a cavity in the right upper lobe (arrow).

**Table 1 medicina-58-01830-t001:** Laboratory findings on admission to our hospital.

	Value	Reference Range
**Complete Blood Count**		
White blood cells	6.08 × 10^9^/L	3.3–8.6 × 10^9^/L
Neutrophils	69%	
Basophils	1%	
Monocytes	8%	
Lymphocytes	22%	
Red blood cells	2.36 × 10^12^/L	3.86–4.92 × 10^12^/L
Hemoglobin	9.1 g/dL	11.6–14.8 g/dL
Hematocrit	27.8%	35.1–44.4%
Mean corpuscular volume	117.8 fL	83.6–98.2 fL
Reticulocytes	2.9 × 10^9^/L	
Platelets	19.5 × 10^9^/L	15.8–34.8 × 10^9^/L
**Coagulation System**		
Prothrombin time	87.3%	70–140%
APTT	24.4 s	25–35 s
**Chemistry**		
Aspartate aminotransferase	14 U/L	13–30 U/L
Alanine aminotransferase	4 U/L	7–23 U/L
Lactate dehydrogenase	212 U/L	124–222 U/L
Total protein	5.5 g/dL	6.6–8.1 g/dL
Albumin	1.6 g/dL	4.1–5.1 g/dL
Globulin	3.9 g/dL	2.2–3.4 g/dL
Total bilirubin	0.2 mg/dL	0.4–1.5 mg/dL
Creatinine	0.32 mg/dL	0.48–0.79 mg/dL
Blood urea nitrogen	5 mg/dL	8–20 mg/dL
Sodium	142 mEq/L	138–145 mEq/L
Potassium	3.7 mEq/L	3.6–4.8 mEq/L
Chloride	105 mEq/L	101–108 mEq/L
Serum ferrum	57 μg/mL	40–188 μg/mL
Ferritin	87 ng/mL	5–152 ng/mL
C-reactive protein	2.41 mg/dL	0.00–0.14 mg/dL
Soluble interleukin-2 receptor	811 U/mL	122–496 U/mL
IgG	308 mg/dL	861–1747 mg/dL
IgA	2072 mg/dL	93–393 mg/dL
IgM	17 mg/dL	50–269 mg/dL
FLC kappa	7.3 mg/L	3.3–19.4 mg/L
FLC lambda	43.8 mg/L	5.7–26.3 mg/L
FLC kappa/lambda ratio	0.17	0.26–1.65
**Infection**		
HIV antibody	negative	
Hepatitis B surface antigen	negative	
Hepatitis B core antibody	negative	
Hepatitis C virus antibody	negative	

APTT: activated partial thromboplastin time, FLC: free light chain, HIV: human immunodeficiency virus, Ig: immunoglobulin.

## Data Availability

All data are included in the main text.

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
