# Peer review of "Intestinal Mucosa-Associated Lymphoid Tissue Lymphoma Transforming into Diffuse Large B-Cell Lymphoma in a Young Adult Patient with Neurofibromatosis Type 1: A Case Report"

_medicina, 2022, doi:10.3390/medicina58121830_

Round 1

Reviewer 1 Report

Thank you for the opportunity to review this case report on Intestinal Mucosa-associated Lymphoid Tissue Lymphoma Transforming into Diffuse Large B-cell Lymphoma in a Young Adult Patient with Neurofibromatosis Type 1. The report describes a very interesting presentation and can be valuable for readers to learn more about. However, there are a few key changes that I would recommend :

The discussion needs improvement. The authors first describe NF1 in detail in the context of this patient’s presentation. However, later in the discussion, there is a mention that this patient may not even meet the diagnostic criteria for NF1. After that, alternative diagnoses like CMMRD and IPSID have been suggested, but those are not entirely fitting with this patient’s clinical features either. At the end of the discussion, the reader is confused about what this patient’s diagnosis truly is and how it is linked in context with the presentation of IgA-expressing MALT lymphoma transforming to a high-grade histology. Please clarify. Did the patient have any other findings that may suggest NF1 such as neurofibromas, axillary/inguinal freckling, lisch nodules, optic glioma, family history etc? If not, this patient cannot be diagnosed with NF1.

There is limited discussion about the rarity of IgA-expressing MALT lymphomas. As this is not a typical or common presentation, it may be correlated with the patient’s pulmonary MALT or GI MALT. The authors should highlight the frequency of histological transformation specifically in IgA-expressing lymphoma. Additionally, I would recommend discussing the possible underlying etiology that resulted in development of MALT as this is commonly associated with underlying infections or inflammatory process. I think that highlighting these points in the discussion could add more value for readers.

Line 49-50: “Histological transformation results in chemotherapy-refractory disease and a worse prognosis.” Please cite a reference supporting this statement.

Author Response

To Reviewer 1:

Thank you for the opportunity to review this case report on Intestinal Mucosa-associated Lymphoid Tissue Lymphoma Transforming into Diffuse Large B-cell Lymphoma in a Young Adult Patient with Neurofibromatosis Type 1. The report describes a very interesting presentation and can be valuable for readers to learn more about. However, there are a few key changes that I would recommend :

> Thank you for reviewing our manuscript kindly. In the revision, we corrected the points that you suggested. The manuscript has significantly improved by incorporating your suggestions. With these changes, we hope that the manuscript is now acceptable for publication. All implemented changes have been highlighted in red in a revised manuscript.

The discussion needs improvement. The authors first describe NF1 in detail in the context of this patient’s presentation. However, later in the discussion, there is a mention that this patient may not even meet the diagnostic criteria for NF1. After that, alternative diagnoses like CMMRD and IPSID have been suggested, but those are not entirely fitting with this patient’s clinical features either. At the end of the discussion, the reader is confused about what this patient’s diagnosis truly is and how it is linked in context with the presentation of IgA- expressing MALT lymphoma transforming to a high-grade histology. Please clarify. Did the patient have any other findings that may suggest NF1 such as neurofibromas, axillary/inguinal freckling, lisch nodules, optic glioma, family history etc? If not, this patient cannot be diagnosed with NF1. (Revised diagnostic criteria for NF1 according to Legius et al.)

> Response: Thank you for your important comment. In the “Case Report” session, we did not include enough information about the diagnosis of NF1. We have added the information that she had axillary freckles (Line 73, Page 2). Our case meets the diagnostic criteria for NF1 according to Legius et al. To clarify them, we have modified the related parts for the readers to easily understand in the “Discussion” session (Line 190-194, Page 6).

There is limited discussion about the rarity of IgA-expressing MALT lymphomas. As this is not a typical or common presentation, it may be correlated with the patient’s pulmonary MALT or GI MALT. The authors should highlight the frequency of histological transformation specifically in IgA-expressing lymphoma.

> Response: Thank you for highlighting the important point. As you suggested, we agree that IgA-expressing MALT lymphoma needs to be discussed. However, IgA-expressing MALT lymphomas are extremely rare. Thus, there has been not enough evidence to discuss the clinical outcome including the histological transformation. We found that one case report of DLBCL transformed from IgA-expressing marginal zone lymphoma (Koyama R, Int J Hematol. 2000; 72: 349-352). We have added the following sentence in the “Discussion” session (Line 231-237, Page 7):

“Although monoclonal gammopathy was found in 15-20% of patients with low-grade and aggressive B-cell lymphoma, the frequency of IgA-expressing lymphomas was very low [34]. The previous retrospective analysis has also described that 1 of 19 patients (5%) with MALT lymphoma had IgA monoclonal gammopathy [35]. Moreover, we found only one case report of histological transformation in IgA-expressing lymphoma [36]. These findings suggest that our patient with histological transformation in IgA-expressing MALT lymphoma was extremely rare.”

[34] Economopoulos T, Leuk Res. 2003 Jun;27(6):505-8. PMID: 12648510

[35] Wöhrer S, Clin Cancer Res. 2004 Nov 1;10(21):7179-81. PMID: 15534090

[36] Koyama R, Int J Hematol. 2000 Oct;72(3):349-52. PMID: 11185993

Additionally, I would recommend discussing the possible underlying etiology that resulted in development of MALT as this is commonly associated with underlying infections or inflammatory process. I think that highlighting these points in the discussion could add more value for readers.

> Response: As you suggested, we have added the following sentence in the “Discussion” session (Line 211-215, Page 7):

MALT lymphomas are often associated with underlying autoimmune disease and chronic viral infection [1-3]. The close examination at the first hospitalization revealed no evidence of these diseases in our patient. Helicobacter pylori infection is the major risk factor associated with the development of gastric MALT lymphoma; however, the pathogenesis of MALT lymphoma of the colon remains unclear [31].”

[31] J Clin Pathol. 2020 Jul;73(7):378-383. PMID: 32034054

Line 49-50: “Histological transformation results in chemotherapy-refractory disease and a worse prognosis.” Please cite a reference supporting this statement.

> Response: As you pointed out, we have added appropriate references and partially corrected the sentence you indicated (Line 49-50, Page 2).

“Histological transformation results in chemotherapy-refractory disease and a worse prognosis.” → “Histological transformation results in a worse prognosis [7-9].”

[7] J Clin Oncol. 2018 Oct 12:JCO1800138. doi: 10.1200/JCO.18.00138. Epub ahead of print. PMID: 30312133.

[8] Br J Haematol. 2019 Aug;186(3):448-459. doi: 10.1111/bjh.15953. Epub 2019 May 24. PMID: 31124124

[9] Cancers (Basel). 2022 Jun 19;14(12):3019. doi: 10.3390/cancers14123019. PMID: 35740684

Reviewer 2 Report

Kosako et al. provide a concise and interesting case report of a MALT lymphoma with histological transformation to DLBCL a young adult patient with NF1. Furthermore, they identified a somatic mutation in the A20 gene (also called TNFAIP3), possibly involved in the development of MALT lymphoma. This study adds data to that hereditary cancer syndrome should be considered as a differential diagnosis in early-onset MALT lymphoma and potential clinical guidance for future similar cases. Presentation of the patient and related discussion are quite appropriate for the scope of the manuscript. I have some minor suggestions that I feel could improve the quality of the manuscript.

1- Previous published case reports on clinicopathologic features of a NHL or HL arising in a NF1 patient can be tabulated.

2- Could you add details of the therapy given and the changes to standard risk therapy in consideration of NF1 and the risk of secondary malignancies. This will be useful for future cases.

3- Please add appropriate reference to sentence below.

-P2 Line 49:  “Histological transformation results in chemotherapy-refractory disease and a worse prognosis.

Author Response

To Reviewer 3:

Kosako et al. provide a concise and interesting case report of a MALT lymphoma with histological transformation to DLBCL a young adult patient with NF1. Furthermore, they identified a somatic mutation in the A20 gene (also called TNFAIP3), possibly involved in the development of MALT lymphoma. This study adds data to that hereditary cancer syndrome should be considered as a differential diagnosis in early-onset MALT lymphoma and potential clinical guidance for future similar cases. Presentation of the patient and related discussion are quite appropriate for the scope of the manuscript. I have some minor suggestions that I feel could improve the quality of the manuscript.

> Thank you for your detailed reading of our manuscript. In the revision, we corrected the points that you suggested. The manuscript has significantly improved by incorporating your suggestions. With these changes, we hope that the manuscript is now acceptable for publication.

1- Previous published case reports on clinicopathologic features of a NHL or HL arising in a NF1 patient can be tabulated.

>Response: In response to your comment, we have carefully re-evaluated lymphoma case reports in patients with NF1, and then found one article with a table (a list) associated with these clinicopathologic features in NF1 patients with lymphomas. Thus, we have added the following sentence in the “Discussion” session (Line 186-188, Page 6): “As for lymphomas in patients with NF1, 44 cases have been reported; 5 cases of Hodgkin lymphoma and 39 cases of non-Hodgkin lymphoma (18 of B cell, 15 of T cell, and 6 of unclassified) [23].

[23] Case Rep Oncol. 2017 Feb;10(1):161-168. PMID: 28413392

2- Could you add details of the therapy given and the changes to standard risk therapy in consideration of NF1 and the risk of secondary malignancies. This will be useful for future cases.

> Response: As you suggested, we have modified the following sentence in the “Case report” section (Line 154, Page 5): Considering the risk of secondary malignancies, we selected at least six courses of R-CHOP (rituximab, cyclophosphamide, doxorubicin, vincristine, prednisolone) as standard therapy for DLBCL, but not followed by radiation therapy or upfront autologous transplantation.

3- Please add appropriate reference to sentence below.

-P2 Line 49: “Histological transformation results in chemotherapy- refractory disease and a worse prognosis.”

> Response: As you pointed out, we have added appropriate references and partially corrected the sentence you indicated (Line 49, Page 2).

“Histological transformation results in chemotherapy-refractory disease and a worse prognosis.” → “Histological transformation results in a worse prognosis [7-9].”

[7] J Clin Oncol. 2018 Oct 12:JCO1800138. doi: 10.1200/JCO.18.00138. Epub ahead of print. PMID: 30312133.

[8] Br J Haematol. 2019 Aug;186(3):448-459. doi: 10.1111/bjh.15953. Epub 2019 May 24. PMID: 31124124

[9] Cancers (Basel). 2022 Jun 19;14(12):3019. doi: 10.3390/cancers14123019. PMID: 35740684
